# Not All That Glitters Is Gold: Attractive Partners Provide Joys and Sorrows

**DOI:** 10.3390/ijerph192013526

**Published:** 2022-10-19

**Authors:** Anna Cardelli, Camilla Matera, Giulia Rosa Policardo, Cristian Di Gesto, Amanda Nerini

**Affiliations:** 1School of Psychology, University of Florence, 50137 Florence, Italy; 2Department of Education, Languages, Intercultures, Literatures and Psychology, University of Florence, 50135 Florence, Italy; 3Department of Health Sciences, University of Florence, 50139 Florence, Italy

**Keywords:** partner attractiveness, partner commentary, body dissatisfaction, cosmetic surgery, relationship quality

## Abstract

Through a 2 × 2 experimental vignette design, we tested if partner perceived attractiveness in interaction with appearance-related comments from one’s partner might affect women and men’s body dissatisfaction, body shame, acceptance of cosmetic surgery, and perceived relationship quality. Participants were 154 women and 157 men living in Italy (mean age = 30.97; all of them were in a couple relationship), who read a vignette describing the purchase of a swimsuit, through which partner attractiveness (poor versus high) and partner commentary (negative versus positive) were manipulated. Some ANCOVAs were performed on women and men separately. For men, partner commentary affected body dissatisfaction with low body fat. Moreover, the main effect of partner attractiveness was found in their consideration of undergoing cosmetic procedures. Among women, a significant partner attractiveness X partner commentary interaction effect emerged on acceptance of cosmetic surgery for social reasons. As regards relationship quality, for women, there was a main effect of partner attractiveness on relational communication, while a marginally significant interaction effect between partner attractiveness and partner commentary emerged for men’s self-disclosure. Our findings suggest that partner attractiveness is generally beneficial, but when combined with negative feedback concerning the appearance, it might lose its advantages. These findings should be considered for planning interventions aimed at both preventing body dissatisfaction and acceptance of cosmetic surgical procedures for not medical reasons and promoting relationship satisfaction among women and men.

## 1. Introduction

Body dissatisfaction, which refers to a subjective discontent for one’s body in general or for some parts of it [1], is chronically widespread among young Western women but is clearly increasing also among men [2,3,4]. Body dissatisfaction is a central factor for developing high levels of anxiety and depression, low self-esteem, and engaging in unhealthy behaviors, such as the use of cosmetic surgery, not for medical reasons [5,6,7]. Body dissatisfaction is also associated with interpersonal outcomes, such as the quality of one’s couple relationship [8].

The perception individuals have of their body, together with the feelings related to it and the behaviors they perform to modify and improve some parts of it, are extremely influenced by a variety of variables, among which social and interpersonal factors play a pivotal role [1,9,10]. Body image is a dynamic construct that derives from attitudes toward one’s physical appearance, fluidly changing in relation to the attractiveness of the individuals in our relational and social context [11] and to the perception that others have of our attractiveness [12].

The present study aimed to examine if individuals in heterosexual couples might evaluate their bodies and the quality of their couple relationship by referring to their partner’s attractiveness. Previous studies have found that partner attractiveness is beneficial for many aspects of couple functioning, such as dyadic adjustment, investment size, and commitment [13,14,15,16,17]. On the other side, it might foster intra-sexual competitiveness and mate retention tactics [18] and increase the salience of appearance within the couple [19,20]. Notably, according to previous research findings, when the partner presents an attribute that is highly desirable, such as a thin body shape, the individual might become at risk of concern about this attribute, thus developing, in this case, weight concern [19,20]. The present study adds to this body of literature by examining, through an experimental vignette design, if and how the perceived attractiveness of one’s partner could influence the perceived quality of one’s couple relationship, one’s body dissatisfaction, body shame levels, and one’s interest in cosmetic surgery as an appearance changing strategy.

### 1.1. Correlates of Partner Attractiveness

Partner attractiveness is a pivotal driver of mate selection. The Ideal Standards Model (ISM) [21] identifies three main dimensions of standards, one of which relates to the domain of passion and attractiveness. Greater consistency between ideal standards and perceptions of a current romantic partner is positively related to relationship satisfaction and psychological well-being [22,23,24]. According to this model and to several empirical findings [13,16,25], being in a relationship with an attractive person is likely to be beneficial for the individual and, therefore, more desirable to maintain. Some studies showed that the more people see their partner as physically attractive, the more they are satisfied within the couple [16,17], although partner attractiveness seems to be more important for men’s than women’s relationship satisfaction [14,15,26,27].

Although partner attractiveness is likely to be beneficial for the quality of one’s couple relationship, there are some circumstances under which having an attractive partner might be threatening for the dyad. This might occur when an attractive partner is supposed to be not committed to the relationship [13]. In this case, having more attractive partners (and/or being faced with intra-sexual competitiveness) might enhance suspicion of extradyadic relationship, jealousy, and mate-guarding behaviors [28,29]. In turn, these behaviors (in particular, jealousy) may be harmful to the relationship and increase relationship conflict [30].

### 1.2. Appearance-Related Comments from a Romantic Partner

Previous research has provided correlational evidence about the association between appearance-related comments from one’s partner and both couple and body satisfaction [31,32].

When appearance-related comments from a romantic partner are not balanced with valuation of nonphysical qualities, the salience of appearance within the couple could increase at the expense of other characteristics [33]. Indeed, according to objectification theory [34], exposure to messages about physical attractiveness encourages individuals to be more aware of their appearance and to see themselves as an object to be looked at and evaluated (self-objectification). This appearance monitoring against unrealistic cultural standards of beauty may result in the adverse psychological consequences of increased body shame, anxiety, and body dissatisfaction [35].

In general, people tend to feel better about their bodies when they perceive that their partners appreciates them [36,37]. Individuals who receive positive rather than negative comments from their partners have a more positive view of their bodies and, consequently, of the partner who expresses these compliments [38]. Conversely, weight criticism between partners is consistently associated with poorer relationship functioning, including lower relationship satisfaction, sexual intimacy, relationship stability, and constructive communication during conflict [39].

Some recent experimental findings confirmed the relationship between the comments received from one’s partner, couple relationship quality, and body image in men and women [40]. In the study by Fornaini and colleagues [40], women who imagined receiving a compliment about their physical appearance from their partners felt more accepted by them and were less afraid of being abandoned or rejected. As for men, imagining being criticized for body weight and shape by the partner caused an increase in body dissatisfaction and a decrease in body compassion (a kind and accepting attitude toward our body), while no effects of the comment on their perceived couple relationship were found.

### 1.3. Partner Attractiveness and Appearance-Related Comments

What does occur when a negative comment is expressed by an attractive partner?

It is possible to assume that attractive individuals who criticize their mates’ appearance might produce negative consequences on their partner’s body image and relationship satisfaction. First, they might enhance their partners’ body dissatisfaction and interest in modifying their bodies by promoting upwards comparison based on physical appearance. According to Social Comparison Theory [41], much research has shown that both women and men who engage more frequently in appearance-related comparison are more likely to feel dissatisfied with their bodies [42,43]. Markey and Markey [20] proposed a model called the Partner Comparison Effect, in which they argued that individuals’ perceptions of their bodies are influenced by a comparison process relative to their partner. This theory was tested with a group of both heterosexual and lesbian women; in line with the authors’ hypotheses, women with a higher Body Mass Index (BMI) who were in couple with thinner partners had a higher risk of perceiving themselves as overweight and were more likely to be dissatisfied with their weight [44].

Partner attractiveness could also influence body image by affecting the perception of one’s partner’s body-related expectations [45,46]. Very attractive people could be supposed to have high levels of expectations of attractiveness in their partners, thus enhancing body-related concerns and body dissatisfaction of the other partner. Reynolds and Meltzer [47] showed that less attractive wives married to more attractive husbands reported more dieting behaviors, which is one of the most common appearance-enhancing strategies among women. Based on these findings, it is reasonable to believe that partner attractiveness could also be associated with a greater interest in other strategies that could help to improve one’s look, such as cosmetic surgery.

Empirical findings showed that cosmetic procedures can be sought by both women and men in order to be more attractive and improve interaction with their romantic partners [48]. Accordingly, research has shown that individuals might accept cosmetic surgery to improve their physical appearance as a tactic for intrasexual competition [49]. Indeed, interest in cosmetic procedures can be due to either intrapersonal reasons, such as body dissatisfaction [50,51] and social reasons, such as improving love dating or marital satisfaction, satisfying their partners or other close relationships, and wanting to be less self-conscious around others [51]. As we stated above, partner attractiveness and negative appearance-related comments could trigger upward appearance-related comparison, which was found to be significantly related not only to body dissatisfaction but also to attitudes towards cosmetic procedures among both women [52] and men [53].

To be criticized for one’s weight or shape by a partner who is attractive might also threaten the quality of one’s couple relationship. Considering the key role of mutual physical appreciation within the dyad [54], it can be predicted that, under these circumstances, individuals will feel more insecure about their relationship. As stated above, an attractive partner who expresses a negative comment appears to judge the other couple member as not appealing or desirable. In such a condition, the relationship could be perceived as at risk, as mate value discrepancy [55] could be perceived as especially high under these circumstances. Attractive people are seen as more able to draw the attention of others and to be appreciated by other women or men; if they do not appreciate the appearance of their mate, which become evident when they express a negative appearance-based comment, they might be perceived as being less committed to the relationship and more interested in finding an alternative mate, who could correspond better to their beauty ideals [56]. To receive negative feedback about one’s appearance from an attractive partner could then increase fears of being abandoned, strengthen individuals’ doubts about their relationship and worsen the quality of the communication within the couple. Additionally, feelings of self-objectification that could be fostered by such a focus on appearance might threaten the quality of one’s couple relationship. Indeed, some recent studies have shown that objectification of one’s partner is related to decreased relationship quality among both women and men [33,57,58,59].

### 1.4. The Present Research

The present study aimed to test if partner attractiveness can affect one’s body image and perceived relationship quality. Although some studies have highlighted an association between partner attractiveness and one’s body image [20] or partner attractiveness and couple relationship [27], only correlational evidence was provided concerning the association between these variables. To the best of our knowledge, there are no studies that have experimentally investigated the role of partner attractiveness on women’s or men’s body image and relationship quality. Through the present research, we aimed to fill this gap by adopting an experimental vignette design, through which we intended to examine if partner attractiveness could moderate the effect that receiving an appearance-related comment from one’s partner could have on one’s body image (i.e., body shame and body dissatisfaction), interest in cosmetic surgery and perceived couple relationship.

We hypothesized that feelings of confidence about either one’s appearance or one’s dyadic relationship could be decreased in individuals who receive a negative comment from a partner who is seen as highly attractive. In such a condition, the negative comment would accentuate the discrepancy between one’s partner’s and one’s own perceived attractiveness, thus promoting upward comparison and increasing feelings of discomfort, shame, and self-body judgment in negative terms. Moreover, having a very attractive partner who expresses appearance-related comments could make the physical appearance salient within the couple, overshadowing other features. Being more likely to view their body in negative and objectified terms, individuals might consider resorting to cosmetic surgery as a strategy to improve their appearance, with the main motivation of resulting more desirable to their partner’s eyes. The effect of a negative comment might be much weaker when people see the partner as poorly attractive; in this case, a negative comment would not accentuate the discrepancy that individuals might perceive between their partners’ and their own attractiveness, thus resulting in being almost uninfluential. Moreover, attractive partners expressing verbal criticism towards their mates might appear not sufficiently committed to the relationship, thus fostering jealousy and relationship conflict. The quality of one’s couple relationship might be less threatened when a negative comment is expressed by a poorly attractive partner, who is less likely to foster jealousy and relational fears or doubts. Notably, partner attractiveness was considered in terms of overall appearance and not just weight. Indeed, general attractiveness is directly related to body dissatisfaction, body dysmorphic disorder, and self-objectification [60].

In sum, we tested the following hypotheses:

**Hypothesis** **1.**
*In the highly attractive partner condition, both women and men exposed to a negative comment about their physical appearance would report higher body dissatisfaction, body shame, and acceptance of cosmetic surgery. In the poorly attractive partner condition, no difference in body dissatisfaction, body shame, and acceptance of cosmetic surgery is expected between participants exposed to a negative or positive comment.*


**Hypothesis** **2.**
*In the highly attractive partner condition, both women and men exposed to a negative comment about their physical appearance would report a lower perception of the quality of their couple relationship. In the poorly attractive partner condition, no difference in couple relationship quality is expected between participants exposed to a negative or positive comment.*


## 2. Materials and Methods

### 2.1. Research Design

Given that most measures for assessing body image are not equivalent for men and women, the hypotheses were tested separately on the two groups. For each group a 2 × 2 experimental between-subject design was adopted. The independent variables were partner attractiveness (Poorly attractive versus Highly attractive) and partner commentary (Negative comment versus Positive comment), which were manipulated by presenting participants with a vignette. In testing our hypotheses, we controlled for participants’ BMI, age, and relationship length.

### 2.2. Participants and Procedure

A power analysis using G*Power [61] indicated that a minimum sample of 128 would be needed to detect medium effects (effect size = 0.25) with 80% power using an ANOVA with 4 groups, 3 covariates, and the alpha at 0.05. At least 128 women and 128 men were then to be recruited.

Overall, 154 women and 157 men took part in the study. Only participants aged between 20 and 50 and within a heterosexual romantic relationship for at least one year were allowed to participate. The female participants had a mean age of 30.97 (*SD* = 9.39), and their average BMI was 21.62 (*SD* = 2.70), ranging from 16.33 to 29.41. Their mean relationship length was around ten years (*M* = 9.45, *SD* = 8.74), from a minimum of 12 months up to a maximum of 34 years. Almost all of the participants were Italian, except one of Swiss nationality. Most of the women lived in Central Italy (96.1%), followed by 2.6% who lived in Northern Italy and 1.3% in Southern Italy. Regarding civil status, 52.6% were unmarried, and 47.4% were married or cohabitating. Regarding education, 42.2% had a high school degree, 29.2% had a bachelor’s degree, 24.7% had a master’s degree, 1.9% had completed lower secondary school, 0.6% had attended elementary school, and 1.3% (2 subjects) reported a different degree (e.g., Ph.D.). With respect to occupation, 47.4% had full-time employment, 11% had part-time employment, 6.5% had occasional work, 0.6% reported that they were looking for their first job, 32.5% were students, and the remaining 1.9% were unemployed.

The male participants had a mean age of 31.59 (*SD* = 8.77), and their average BMI was 24.03 (*SD* = 3.25), ranging from 17.72 to 36.02. Their mean relationship length was around 8 years (*M* = 7.98, *SD* = 7.43), from a minimum of 12 months up to a maximum of 32 years. All of the participants were Italian. Most of them lived in Central Italy (98.1%), followed by 1.9% who lived in Northern Italy. Regarding civil status, 59.2% were unmarried, 40.1% were married or cohabitating, and only 1 person was divorced. Regarding education, 48.4% had a high school degree, 23.6% had a bachelor’s degree, 21% had a master’s degree, 5.7% had completed lower secondary school, and 2 participants (1.3%) reported a different degree (e.g., Ph.D.). With respect to occupation, 61.8% had full-time employment, 8.9% had part-time employment, 3.2% had occasional work, 1.9% reported that they were looking for their first job, 22.9% were students, and the remaining 1.3% were unemployed.

The study participants were recruited using opportunistic sampling techniques in order to reach the required sample size in a short time and avoid the risk of incurring possible restrictions due to the progress of the COVID-19 pandemic. We asked the participants to take part in a study on body image and the quality of couple relationship. Participation in the study was voluntary, and incentives were not provided. The participants gave their informed consent prior to completing the questionnaire, and their responses were recorded anonymously in full compliance with the privacy regulations. Each participant completed the questionnaire alone and gave it back to the researcher. On average, each participant took 20 min to complete the questionnaire. Keeping men and women separately, the participants were randomly assigned to one of the four experimental conditions (Positive comment and Highly attractive partner, Positive comment and Poorly attractive partner, Negative comment and Highly attractive partner, Negative comment and Poorly attractive partner). The number of men and women in each group is reported in Table 1 and Table 2. After completing the questionnaire, each respondent was carefully debriefed. The study procedure was approved by the Ethical Committee of the university with which the authors are affiliated (Prot. 0037781, 1 February 2021).

### 2.3. Measures

All of the participants read a vignette through which the independent variables were manipulated. The vignette described the purchase of a swimsuit (bikini for women and speedos for men) along with one’s partner, which might elicit a state of activation and self-awareness towards one’s body. Previous studies showed that for both women and men, wearing a swimsuit is an uncomfortable and distressing body-related situation that produces negative outcomes in terms of self-esteem and self-objectification [34,62,63,64]. The participants were asked to read the vignette carefully and calmly, imagining themselves in the described situation. For the purpose of our study, the vignettes previously created by Fornaini et al. [40] were partially modified. Participants were asked to imagine that summer was coming; they and their partners had booked a week vacation in a seaside resort. They decided to go shopping together, as they needed a new swimsuit. They entered a shop, and after looking around, they both saw a swimsuit that they liked. They went to the dressing room and took a turn trying it on. When their partners tried the swimsuit on, the participants imagined observing them and thinking they had a beautiful body and the swimsuit fitted them very well (Highly attractive partner condition) or thinking that they did not have a beautiful body and that the swimsuit did not fit them at all (Poorly attractive partner condition). When participants imagined themselves in a swimsuit, they also imagined receiving a comment from the partner; in the positive comment condition, the partner expressed some appearance-related compliments, while in the negative comment condition, the partner expressed some criticism about the partner’s body weight and shape (see Appendix A).

The order of presentation of the partner commentary and the partner attractiveness was counterbalanced so that half participants read the vignette in which the commentary was manipulated before the partner attractiveness, and half participants read the vignette in which the partner attractiveness was manipulated before the commentary. No significant order effects were found either for men or for women. In line with previous studies [40], we preferred not to use a pre-test assessing body image in order to avoid any influence on participants’ perception of the situation described in the vignette (for methodological consideration, see Pasnak [65]. After reading the vignette, the participants completed items and scales aimed to assess the following variables.

*Partner attractiveness manipulation check*. One single item was used to check for the manipulation of partner attractiveness. Participants were asked to indicate how they evaluated the attractiveness of their partner, providing their responses on a 9-point Likert scale from (1 = completely negative evaluation; 9 = completely positive evaluation).

*Commentary manipulation check*. An item asking the participants to judge the comment they had received constituted our commentary manipulation check. Participants provided their responses on a 9-point Likert scale from (1 = completely negative; 9 = completely positive).

*Women’s Body dissatisfaction*. Women’s body dissatisfaction was measured through the Italian version [66] of the *Body Shape Questionnaire-14* (BSQ-14) [67], which is composed of 14 items (e.g., “Has seeing your reflection—e.g., in a mirror or shop window—made you feel bad about your shape?”) with a 6-point Likert-type response format (1 = never; 6 = always). Higher scores indicate higher levels of body dissatisfaction. Instructions were slightly modified so that participants were asked to think about how they felt about their bodies at that moment (state body dissatisfaction). In the present study, the internal consistency of the scale was very good (𝛼 = 0.92).

*Men’s Body dissatisfaction*. The Italian version [68] of the *Male Body Attitudes Scale* (MBAS) [69] was used to assess men’s body dissatisfaction. This scale is composed of 24 items with a 6-point Likert-type response format (1 = never; 6 = always). Three subscales comprise the MBAS, namely *Muscularity* (10 items, e.g., “I think my chest should be larger and more defined”), which measures men’s attitudes toward their muscularity; low *Body Fat* (8 items, e.g., “I think I have too much fat on my body”), which assesses men’s attitudes toward their body fat, and Height (2 items, e.g., “I wish I were taller”), which measures men’s attitudes toward their height. Higher scores indicate higher body dissatisfaction. In the present study, the Cronbach’s alpha for the subscales were the following: *Muscularity* = 0.83; *Low Body Fat* = 0.91; Height = 0.80.

*Body shame*. Body shame was assessed through the Body Shame subscale of the Italian version [70] of the *Objectified Body Consciousness Scale* (OBCS) [71]. The *Body Shame* subscale assesses shame felt due to one’s body not fitting society’s expectations (e.g., “When I don’t have the size I think I should have, I feel ashamed”). This subscale is composed of 8 items with a 7-point Likert scale (from 1= strongly disagree to 7= strongly agree). Higher scores indicate a higher feeling of shame toward one’s body. In the present study, Cronbach’s alpha for this subscale was good (Women: 0.80; Men: 0.75).

*Attitudes towards cosmetic surgery*. Attitudes towards cosmetic surgery were measured through the Italian Version [72] of the *Acceptance of Cosmetic Surgery Scale* (ACSS) [51]. The ACSS is composed of 15 items, with a 7-point Likert-type response format (1 = strongly disagree; 7 = strongly agree). Three subscales comprise the ACSS. The *Intrapersonal* subscale measures the degree to which cosmetic surgery is believed to offer intrapsychic benefits or self-image improvements (5 items; e.g., “It makes sense to have small cosmetic surgical treatments instead of spending years feeling uncomfortable about the way you look”); the *Social* subscale measures beliefs about the social or interpersonal benefits offered by cosmetic surgery (5 items; e.g., “If a simple cosmetic surgery procedure made me more attractive to my partner, I would think about the possibility of trying it”); the *Consider* subscale measures the probability that an individual has to undergo cosmetic surgery (5 items; e.g., “If I knew there are none negative side effects or pain, I would like to try cosmetic surgery”). In the present study the Cronbach’s 𝛼 of the scales were the following: Women: *Intrapersonal* = 0.92, *Social* = 0.80, *Consider* = 0.88; Men: *Intrapersonal* = 0.85, *Social* = 0.86, *Consider* = 0.84.

*Quality of couple relationship*. Relationship affectivity was measured through five subscales of the *Couple’s Affectivity Scale* (CAS) [73], which is a multidimensional measure assessing several aspects of a couple’s relationship. Items have a 5-point Likert-type response format (1 = never; 5 = always). The CAS was developed and validated with Italian women and men. In the present study, we used the following subscales of the CAS: *Self-disclosure* (SD; 4 items, e.g., “You have expressed to your partner unpleasant events that occurred during the day”) to measure willingness and openness to dialogue with a partner; *Partner-disclosure* (PD; 5 items, e.g., “Your partner has told you about his feelings”) to assess the respondents’ perception of their partner’s openness to expressing ideas and feelings; *Perceived Partner Responsiveness* (PPR; 5 items, e.g., “Your partner has shown that he esteems you”) to measure the respondents’ perception of the ability of their partner to express understanding, affection, esteem, and support; *Relational Communication* (RC; 3 items, e.g., “You have talked with your partner about your relationship”) to assess the ability to talk about the relationship and address couple problems; and *Fears of being Abandoned and Rejected* (FAR; 3 items, e.g., “You have happened to be afraid of loneliness”) to measure the fear of being abandoned and not accepted by the partner. The Cronbach’s 𝛼 of the subscales were the following: Women: SD = 0.79; PD = 0.70; PPR = 0.77; RC = 0.68; FAR = 0.67; Men: SD = 0.66; PD = 0.61; PPR = 0.80; RC = 0.63; FAR = 0.70.

Each participant reported sex, nationality, place of residence, age, the length of their relationship (expressed in months), civil status, educational level, and occupational status. We calculated BMI_s_ (kg/m^2^) using the participants’ reported weights and heights.

## 3. Results

Descriptive statistics are reported in Table 1 and Table 2. As specified above, all the analyses were conducted separately for women and men. First of all, we performed a series of ANOVAs in order to test if the participants differed for key variables potentially associated with body image or relationship satisfaction (BMI, age, relationship length) based on the experimental condition to which they were assigned. A main effect of the condition did not emerge for any of the variables (all *p* > 0.13). Thus, randomization resulted in equivalent groups on these variables.

**Table 1 ijerph-19-13526-t001:** Means (SDs) of body dissatisfaction, acceptance of cosmetic surgery, and quality of couple’s relationship by experimental condition (men).

Partner Commentary	Partner Attractiveness		Comm. Check	Partner Attr. Check	MBAS LBF	MBAS Musc.	MBAS Height	Body Shame	ACSS Cons.	ACSS Intra.	ACCS Soc.	CAS SD	CAS PD	CAS PPR	CAS RC	CAS FAR
		N	*M* *(SD)*	*M* *(SD)*	*M* *(SD)*	*M* *(SD)*	*M* *(SD)*	*M* *(SD)*	*M* *(SD)*	*M* *(SD)*	*M* *(SD)*	*M* *(SD)*	*M* *(SD)*	*M* *(SD)*	*M* *(SD)*	*M* *(SD)*
Negative	Poor	36	3.44 (1.78)	3.44 (1.81)	2.93 (1.28)	2.97 (0.94)	2.81 (1.49)	2.74 (1.00)	2.46 (1.60)	3.79 (1.60)	1.77 (1.13)	2.90 (0.71)	2.94 (0.53)	3.19 (0.57)	2.54 (0.97)	1.31 (0.96)
	High	47	3.60 (1.48)	7.72 (0.99)	2.85 (1.26)	2.72 (0.89)	2.59 (1.49)	2.64 (0.81)	2.13 (1.16)	3.32 (1.27)	1.62 (.85)	2.72 (0.62)	2.92 (0.53)	3.23 (0.54)	2.32 (0.75)	0.90 (0.85)
Positive	Poor	38	7.42 (1.46)	3.08 (1.62)	2.82 (1.29)	3.01 (0.93)	2.78 (1.56)	3.01 (1.18)	2.23 (1.51)	3.85 (1.67)	1.82 (1.08)	2.76 (0.74)	2.67 (0.56)	2.94 (0.79)	2.39 (0.97)	0.89 (0.84)
	High	36	7.78 (1.51)	8.06 (1.01)	2.57 (1.10)	2.73 (0.94)	2.81 (1.55)	2.71 (0.99)	1.80 (1.09)	3.77 (1.36)	1.51 (1.03)	2.98 (0.62)	2.86 (0.55)	3.19 (0.68)	2.29 (0.83)	0.94 (0.84)

**Table 2 ijerph-19-13526-t002:** Means (SDs) of body dissatisfaction, acceptance of cosmetic surgery, and quality of couple’s relationship by experimental condition (women).

Partner Commentary	Partner Attractiveness		Comm. Check	Partner Attr. Check	BSQ-14	Body Shame	ACSS Cons.	ACSS Intra.	ACCS Soc.	CAS SD	CAS PD	CAS PPR	CAS RC	CAS FAR
		N	*M* *(SD)*	*M* *(SD)*	*M* *(SD)*	*M* *(SD)*	*M* *(SD)*	*M* *(SD)*	*M* *(SD)*	*M* *(SD)*	*M* *(SD)*	*M* *(SD)*	*M* *(SD)*	*M* *(SD)*
Negative	Poor	36	2.97 (1.83)	3.28 (1.85)	2.50 (0.92)	3.00 (1.02)	2.65 (1.69)	4.02 (1.77)	1.27 (0.54)	3.11 (0.62)	2.43 (0.61)	2.97 (0.55)	2.06 (0.76)	0.94 (0.80)
	High	43	2.07 (1.40)	8.12 (1.22)	2.91 (0.94)	3.41 (1.45)	3.12 (1.59)	4.22 (1.52)	1.62 (0.74)	2.99 (0.77)	2.62 (0.72)	3.15 (0.71)	2.44 (0.84)	1.12 (0.88)
Positive	Poor	35	8.29 (0.96)	3.23 (1.66)	2.57 (0.88)	3.08 (1.11)	2.90 (1.86)	3.89 (1.56)	1.63 (0.93)	3.19 (0.64)	2.46 (0.83)	3.13 (0.63)	2.30 (1.00)	0.89 (0.71)
	High	40	8.08 (1.10)	8.15 (0.98)	2.84 (1.11)	3.30 (1.43)	2.71 (1.63)	4.20 (1.67)	1.49 (0.74)	3.14 (0.47)	2.58 (0.63)	3.23 (0.61)	2.69 (0.85)	1.03 (0.90)

We then controlled for the effectiveness of the manipulations among both women and men. For women, an ANOVA revealed a main effect of Partner attractiveness on the Partner attractiveness manipulation check (*F*_(1,150)_ = 436.52, *p* < 0.001, *η^2^* = 0.74); in the Highly attractive partner condition, the partner was evaluated as more attractive (*M* = 8.13) than in the Poorly attractive partner condition (*M* = 3.25). An ANOVA revealed also a main effect of the Partner commentary on the commentary manipulation check (*F_(_*_1,150)_ = 661.59, *p* < 0.001, *η^2^* = 0.81). In line with our predictions, in the Positive comment condition, participants rated the commentary more positively (*M* = 8.18) than they did in the Negative comment condition (*M* = 2.52). Additionally, a main effect of Partner attractiveness emerged on this manipulation check (*F*_(1,150)_ = 6.40, *p* < 0.05, *η^2^* = 0.04). For women, the comment was evaluated as more positive when it came from a Poorly attractive (*M* = 5.63) rather than a Highly attractive (*M* = 5.07) partner.

For men, an ANOVA revealed a main effect of Partner attractiveness on the Partner attractiveness manipulation check (*F*_(1,153)_ = 433.76, *p* < 0.001, *η^2^* = 0.74); in the Highly attractive partner condition, the partner was evaluated as more attractive (*M* = 7.89) than in the Poorly attractive partner condition (*M* = 3.26). An ANOVA revealed also a main effect of Partner commentary on the commentary manipulation check (*F*_(1,153)_ = 265.63, *p* < 0.001, *η^2^* = 0.63). In the Positive comment condition, participants rated the commentary more positively (*M* = 7.60) than they did in the Negative comment condition (*M* = 3.52).

Correlations among the research variables are reported in Table 3 and Table 4. Age, relationship satisfaction, and BMI were significantly correlated with some of the criterion variables among both women and men.

Based on these findings, these variables were controlled for in the subsequent analyses. To test the hypotheses, a series of ANCOVAs were conducted, including Partner commentary and Partner attractiveness as the independent variables; age, relationship length, and BMI were included as the covariates; body dissatisfaction, acceptance of cosmetic surgery, and quality of one’s couple relationship were respectively the dependent variables.

As regards body image, contrary to the first hypothesis, we did not find a Partner attractiveness x Partner commentary interaction effect on women’s either body dissatisfaction or body shame. For both body dissatisfaction, *F*_(1,147)_ = 50.60, *p* < 0.001, *η^2^* = 0.26, and body shame, *F*_(1,147)_ = 14.54, *p* < 0.001, *η^2^* = 0.09, the effect of the covariate BMI was significant. For men, a main effect of the Partner commentary on body dissatisfaction with Low Body Fat emerged, *F*_(1,151)_ = 4.55, *p* < 0.05, *η^2^* = 0.03. Participants who received the negative comment reported greater body dissatisfaction with body fat (*M* = 2.95) than those who received the positive comment (*M* = 2.63). The effect of the covariate BMI was significant, *F*_(1,151)_ = 104.13, *p* < 0.001, *η^2^* = 0.41. Only significant effects of BMI (*F*_(1,150)_ = 4.67, *p* < 0.05, *η^2^* = 0.03) and age (*F*_(1,150)_ = 6.12, *p* < 0.05, *η^2^* = 0.04) emerged on men’s dissatisfaction with their muscularity, while no significant effects of either the independent variables or the covariates were found on men’s dissatisfaction with their height. As regards men’s body shame, only the effect of the covariate BMI was significant, *F*_(1,150)_ = 12.48, *p* < 0.001, *η^2^* = 0.08.

With respect to acceptance of cosmetic surgery, in line with our hypothesis, a significant Partner attractiveness X Partner commentary interaction effect emerged on women’s acceptance of cosmetic surgery for social reasons, *F*_(1,147)_ = 4.00, *p* < 0.05, *η^2^* = 0.03; simple effects analysis revealed that women were more interested in cosmetic surgery for social issues when they imagined receiving a negative comment from a highly attractive partner (*M* = 1.63) than from a poorly attractive one (*M* = 1.28). In the positive commentary condition, no differences emerged between the Highly (*M* = 1.47) and Poorly attractive partner conditions (*M* = 1.62) with respect to respondents’ interest in cosmetic surgery for social reasons. No significant effect emerged for male participants on the Social subscale. With respect to the Consider subscale, among women only a significant effect of the covariates age (*F*_(1,147)_ = 4.29, *p* < 0.05, *η^2^* = 0.03) and relationship length (*F*_(1,147)_ = 6.49, *p* < 0.05, *η^2^* = 0.04) emerged. For men, a main effect of Partner attractiveness was found, *F*_(1,150)_ = 3.92, *p* = 0.05, *η^2^* = 0.03. Consideration of cosmetic surgery was higher among men when they imagined having a poorly (*M* = 2.36) rather than a highly (*M* = 1.94) attractive partner. For the Intrapersonal subscale, no significant effects of our independent variables and covariates were found for either women or men.

We then tested the second hypothesis and examined the effect of the independent variables on participants’ reported quality of their couple relationship. For women, the ANCOVA revealed a main effect of Partner attractiveness on one of the subscales of the CAS, namely the Relational communication subscale, *F*_(1,147)_ = 7.15, *p* < 0.01, *η^2^* = 0.05. In the Highly attractive partner condition, respondents perceived greater relational communication within their couple (*M* = 2.57) compared to participants in the Poorly attractive partner condition (*M* = 2.18).

A marginally significant interaction effect between Partner attractiveness and Partner commentary emerged for men’s self-disclosure, *F*_(1,150)_ = 3.59, *p* = 0.06, *η^2^*= 0.02. In the Highly attractive partner condition, self-disclosure was lower when male participants imagined receiving a negative (*M* = 2.73) rather than a positive (*M* = 2.99) comment from their partner. No difference based on the valance of the commentary emerged in the Poorly attractive partner condition. For the PPR subscale, there was a significant effect of the covariate BMI for women, *F*_(1,147)_ = 4.91, *p* < 0.05, *η^2^*= 0.03. No other significant effects were found on the other subscales of the CAS for either women or men.

## 4. Discussion

Building on previous research on the role of romantic partners, the present study examined if and how partner attractiveness might affect body image and perceived relationship quality. We were interested in understanding if perceiving one’s partner as more or less appealing could moderate the effect of receiving a negative appearance-related comment from him/her.

Contrary to Hypothesis 1, imagining having a relationship with a partner evaluated as physically attractive did not affect either women’s or men’s body dissatisfaction and body shame. This could be due to the fact that we manipulated attractiveness by considering overall appearance, but we measured dissatisfaction with weight and shape, not with one’s general body. Participants’ BMI was associated with both women’s and men’s body dissatisfaction and body shame, which suggests that thin ideals are internalized by the two genders. Nevertheless, we should notice that among our female participants, the maximum BMI value was at the upper limit of the overweight category, while among men, the maximum value was at the lower limit of the extremely obese category, which could have affected our results.

In line with a previous experimental study conducted by Fornaini and colleagues [40], a significant role of the comment received from one’s partner on men’s dissatisfaction with low body fat was found; men receiving a negative comment reported greater levels of body dissatisfaction with low body fat than those receiving a positive comment. It seems that being disapproved by one’s partner is harmful to men’s satisfaction with their bodies, regardless of whether their partner is perceived as attractive or not. Only dissatisfaction with body fat was affected by the comment participants imagined receiving, probably because this comment was particularly focused on weight and shape [40]. Body shame, which is a component of self-objectification, was not affected by our manipulations. Maybe the situations the participants were asked to imagine led all of them to feel objectified, independently of the condition to which they were allocated. Indeed, the purchase of a swimsuit has been described as an objectifying condition in which self-awareness towards one’s body can be especially high [62,74]. Moreover, some experimental evidence revealed that even a positive compliment about one’s appearance, which makes participants feel good in general, might paradoxically lead to increased body shame, as it contributes to focusing attention on the external [75].

Even though partner attractiveness was not found to affect participants’ dissatisfaction with their bodies, it did affect both men’s and women’s acceptance of cosmetic procedures, with some interesting gender differences. In line with the first hypothesis, women who imagined receiving a negative comment from an attractive partner resulted in being more interested in cosmetic procedures for social reasons, among which being more attractive to one’s mate might be especially important. In such a condition, cosmetic surgery might be seen as a strategy that could keep the partner close: given that he is perceived as physically attractive, other women might be interested in him, threatening the stability of the relationship. For these women, cosmetic surgical procedures could be seen as a way to become more appealing to their partners and improve their interaction with them [48]. These findings are in line with previous research indicating that women might consider cosmetic surgery as a benefit-provisioning strategy to retain their mate [76] and succeed in intrasexual competition [49], as their mate values highly depend on physical appearance [77].

Men’s likelihood to undergo cosmetic procedures was influenced by their partner’s attractiveness, even though men were more likely to consider undergoing some cosmetic surgical procedures when their partner was perceived as poor rather than highly attractive. This finding could be interpreted by referring to the relevance that partner attractiveness has for men. Given that men consider it particularly important to have an appealing partner [46], we could suppose that appearance becomes especially salient to men when the partner is perceived as unattractive rather than attractive. Having an attractive partner could be perceived as somewhat normative, so imagining having a poorly attractive partner might direct men’s attention to appearance more than imagining having a good-looking partner. Being appearance more salient in this condition, men might become especially aware of their aspect and focus more on it; increased self-awareness could thus enhance their consideration of cosmetic surgery as a strategy to improve their physical features [78]. Such a consideration might be independent of their dissatisfaction with body fat or muscularity, as we know that men’s interest in cosmetic procedures is not necessarily related to the extent to which they are satisfied with their body weight and shape [53]. These results suggest that having an attractive partner might be protective against men’s consideration of undergoing surgical procedures for merely cosmetic reasons.

Having an attractive partner was also associated with positive relational outcomes among women. Indeed, the study revealed a significant effect of partner attractiveness on women’s relational communication. When women judged their partner as highly attractive, they reported greater relationship communication than in the poorly attractive partner condition. These findings confirm that having an attractive partner is beneficial for the quality of the relationship, at least what concerns relational communication. Imagining having an attractive partner leads women to perceive it easier to talk with him about their relationship, problems, and relational expectations. The main effect of partner attractiveness was not found in men’s reported quality of their relationship.

Partially in line with Hypothesis 2, a marginally significant interaction effect between partner attractiveness and partner commentary was found for men’s self-disclosure. Couple relationship, in terms of self-disclosure, was lower when an attractive partner expressed a negative appearance-related commentary. If men perceive to be not appreciated by a highly attractive partner, they might think that the partner has better relational alternatives and could be less committed to the relationship. Such a perception might determine a detachment on the part of men, which is expressed in greater coldness and less willingness to communicate and dialogue with their partner. Contrary to our hypothesis, the combined effect of partner attractiveness and partner commentary did not affect any aspect of women’s perceived couple relationship.

This study represents one of the first attempts to investigate the effects (on body image and couple relationship quality) of partner attractiveness and its interaction with the appearance-related comments that the partner could express. Our findings suggest that partner attractiveness is generally beneficial for both women and men. Imagining having an attractive partner did not affect men’s and women’s body dissatisfaction and body shame; it was positively related to relational communication among women and to lower consideration of cosmetic surgery among men. Nevertheless, when combined with negative feedback concerning the appearance, partner attractiveness might lose its advantages. Indeed, imagining having an attractive partner who disapproves of the other couple member’s appearance was associated with higher consideration of cosmetic surgery for social reasons among women and with lower intention to communicate personal information to their mate among men.

The greatest strength of this research is that it has adopted an experimental design, which allows it to determine cause–effect relationships between variables. Nevertheless, we should acknowledge that partner attractiveness was manipulated through a vignette representing a situation in which the participants were called to identify themselves. Although the use of this technique was shown to be effective, the actual attractiveness of the participants’ partner was not considered; if the attractiveness of their actual partner in their real life was very discordant from the one represented in the scenario they read, identification could have been difficult to realize. Likewise, the comments were not directly received from one’s partner but, instead, transmitted through the vignette in which participants were asked to identify; imaging to receive a comment may be different from receiving it in everyday life. Additionally, with regard to many of the effects that were significant, their power was low; this could be due to the fact that the participants were not really living the experience but were only called to imagine it. We should also notice that although both partner attractiveness and partner commentary were successfully manipulated, they were somewhat confounded for women, as a small but significant interaction effect between the two variables was found: women tended to rate more positively a comment received by a partner who they considered poorly rather than highly attractive. It is, therefore, not easy to perfectly disentangle the effects of the two variables, which could partially influence each other. Another limitation is represented by our sampling technique, which was an opportunistic one. Moreover, although we did not recruit couples, we did not control if both partners took part in the study; although unlikely, this possibility cannot be ruled out. Finally, in this study, we did not consider potential differences due to demographic variables, such as living in large multicultural urban settlements or in small monocultural towns or villages, which could affect both one’s body image and one’s couple life.

For what concerns future research, the study could be replicated with individuals in a same-sex couple relationship to check for differences based on sexual orientation. Future studies could also experimentally investigate reciprocal influences between partners. This might be possible by recruiting both members of the couple and assigning them to the same experimental condition, with a vignette describing a dialogue between the partners, in which both receive a comment and express an evaluation of the other. It would be interesting to investigate the effect of partner attractiveness and partner commentary on sexual satisfaction since previous studies have rarely focused on this aspect, which seems to act as a mediator in the relationship between body satisfaction and relational outcomes [8,79,80]. Finally, future studies could examine if partner attractiveness, together with appearance-related comments from one’s partner, could influence men’s and women’s either positive body image, which is receiving increasing attention in recent years [81] or some other appearance-enhancing strategies, such as dieting or compulsive exercise. The role of self-esteem could be considered as well because appearance-related comments could have different effects if they match or mismatch one’s self-view (e.g., positive versus negative comment directed to an individual with high versus low self-esteem).

## 5. Conclusions

These findings have relevant practical implications. First of all, interventions aimed at improving one’s body image might usefully consider the role of the partner, especially for men, who seemed to be especially vulnerable to the negative comments they could receive. Second, women’s motivations for considering not medically necessary surgical procedures should be carefully examined in order to understand if cosmetic surgery is merely sought to please one’s partner. In this vein, counseling could help women to understand if their choice is really driven by intrinsic motivation. The same can be said for men that could decide to undergo cosmetic surgery because of their partner’s attractiveness levels rather than for more personal and stable reasons. The role of one’s partner’s attractiveness, together with communication styles within the couple, could be considered in interventions aimed at improving couple satisfaction and stability.

## Figures and Tables

**Table 3 ijerph-19-13526-t003:** Intercorrelations between variables (men).

	1	2	3	4	5	6	7	8	9	10	11	12	13	14
1. Age														
2. BMI	0.33 ***													
3. Relationship length	0.71 ***	0.17 *												
4. MBAS LBF	0.21 **	0.65 ***	0.12											
5. MBAS muscularity	−0.30 ***	−0.25 **	−0.16 *	0.12										
6. MBAS height	−0.06	0.01	0.02	0.04	0.19 *									
7. Body shame	0.17 *	0.32 ***	0.15	0.51 ***	0.21 **	0.08								
8. ACSS intrapersonal	0.11	0.05	0.08	0.01	0.09	0.13	0.12							
9. ACSS social	0.09	0.13	0.08	0.21 **	0.16 *	0.13	0.27 ***	0.38 ***						
10. ACSS consider	0.18 *	0.12	0.10	0.25 **	0.12	0.02	0.36 ***	0.47 ***	0.63 ***					
11. SD	−0.25 **	−0.07	−0.27 ***	−0.10	−0.03	−0.08	−0.10	−0.08	0.02	−0.07				
12. PD	−0.12	−0.20 **	0.02	−0.21 **	−0.07	−0.06	−0.27 ***	−0.05	−0.14	−0.14	0.48 ***			
13. PPR	−0.18 *	−0.12	−0.15	−0.20 *	−0.21 **	−0.24 **	−0.26 ***	−0.13	−0.18 *	−0.13	0.46 ***	0.55 ***		
14. RC	−0.17 *	−0.13	−0.11	−0.14	−0.03	−0.06	−0.04	−0.02	0.04	−0.01	0.34 ***	0.42 ***	0.32 ***	
15. FAR	0.07	0.05	0.02	0.21 **	0.12	0.02	0.45 ***	0.06	0.20*	0.13	−0.01	−0.08	−0.25 **	0.14

* *p* < 0.05; ** *p* < 0.01; *** *p* < 0.001.

**Table 4 ijerph-19-13526-t004:** Intercorrelations between variables (women).

	1	2	3	4	5	6	7	8	9	10	11	12
1. Age												
2. BMI	0.31 ***											
3. Relationship length	0.82 ***	0.32 ***										
4. BSQ-14	−0.08	0.46 ***	−0.07									
5. Body shame	−0.05	0.27 ***	−0.09	0.78 ***								
6. ACSS intrapersonal	−0.08	−0.04	−0.07	−0.03	0.01							
7. ACSS social	−0.01	0.02	−0.05	0.06	0.13	0.37 ***						
8. ACSS consider	0.01	0.03	−0.10	0.16 *	0.21 **	0.57 ***	0.56 ***					
9. SD	−0.08	0.04	−0.06	0.09	0.04	−0.04	−0.19 *	−0.12				
10. PD	−0.13	0.05	−0.11	0.10	0.02	0.09	0.02	0.01	0.47 ***			
11. PPR	−0.08	0.16 *	−0.07	0.14	0.06	0.07	−0.05	−0.03	0.46 ***	0.61 ***		
12. RC	0.08	0.12	0.02	0.13	0.11	−0.10	−0.14	−0.06	0.42 ***	0.23 **	0.26 **	
13. FAR	−0.24 **	−0.12	−0.26 **	0.12	0.17 *	−0.06	0.06	0.01	−0.01	0.05	0.01	0.20 *

* *p* < 0.05; ** *p* < 0.01; *** *p* < 0.001.

## Data Availability

The data that support the findings of this study are available on request from the corresponding author. The data are not publicly available due to their containing information that could compromise the privacy of research participants. However, should anyone need to access a specific part of the datasets, we will do our best to comply with their request.

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
