# Peer review of "Not All That Glitters Is Gold: Attractive Partners Provide Joys and Sorrows"

_ijerph, 2022, doi:10.3390/ijerph192013526_

Round 1

Reviewer 1 Report

The article is interesting, well-written, and builds upon the interest of the authors in the topic. By comparison to previous articles authored by the research team it brings to attention the topic of acceptance of cosmetic surgery as a means to maintain the quality of a romantic/sexual relationship. The experimental vignette study is presented convincingly. However, there are minor issues that I believe need addressing.

1. The title proposes "joys and sorrows" in the relationship, while in the discussion of the results the positive part is not very prominent. 

2. In the Abstract the wording "unnecessary cosmetic surgical procedures" indicates a negative attitude of the authors towards the topic. Later in the article it becomes clear that the cosmetic surgery is envisaged as a response to a social pressure, but a more neutral presentation should be done. Also, the article refers to Western people (L 24). It would be useful to find this information also in the abstract.

3. In the Method section, participants are described according to numerous features. Are they all from large urban settlements, or small settlements (small tows, villages) also represented here? Is there a difference? Is there a difference between people of different lifestyles? Living in mono-cultural environments or in multicultural settings? 

4. I would have liked to see also the comments proposed in the vignette: compliment, assessment, neutral comment, critical remark etc. (maybe in an Appendix? A sample of the vignette for women/men?)

5. A matter of style: the authors use "we"/"our" too much in the article. When "we"/"our" refers to the research team (e.g. L 41-43) it is acceptable. However when "we"/"our" is used to include also the reader (as in L 34-40 or L 100 "our body") it makes the assumption that there is a consensus regarding the presented issue. I recommend throughout the article the use of a more neutral/scientific style: one, they, it etc.

Author Response

Reviewer #1:

(1) The title proposes "joys and sorrows" in the relationship, while in the discussion of the results the positive part is not very prominent.

Thank you for this comment. In the discussion we incorporated come more comments about the beneficial outcomes (“joys”) of having an attractive partner p.10 we reported that “These results suggest that having an attractive partner might be protective against men’s consideration of undergoing surgical procedures for mere cosmetic reasons. Having an attractive partner was also associated with positive relational outcomes among women”. p.11 we stated that “Our findings suggest that partner attractiveness is generally beneficial for both women and men”.

(2) In the Abstract the wording "unnecessary cosmetic surgical procedures" indicates a negative attitude of the authors towards the topic. Later in the article it becomes clear that the cosmetic surgery is envisaged as a response to a social pressure, but a more neutral presentation should be done.

Thank you, we acknowledge that a more neutral presentation of cosmetic surgery should be done in the abstract. Consequently, we modified the sentence as follows: “These findings should be considered for planning interventions aimed at both preventing body dissatisfaction and acceptance of cosmetic surgical procedures for not medical reasons”.

(3) Also, the article refers to Western people (L 24). It would be useful to find this information also in the abstract.

As suggested, in the abstract we specified that participants were living in Italy.

(4) In the Method section, participants are described according to numerous features. Are they all from large urban settlements, or small settlements (small tows, villages) also represented here? Is there a difference? Is there a difference between people of different lifestyles? Living in mono-cultural environments or in multicultural settings? 

Unfortunately, we did not collect this information. We added this as a limitation of our study (“In this study we did not consider potential differences due to demographic variables, such as living in large multicultural urban settlements or in small monocultural towns or villages, which could affect both one’s body image and one’s couple life”).

(5) I would have liked to see also the comments proposed in the vignette: compliment, assessment, neutral comment, critical remark etc. (maybe in an Appendix? A sample of the vignette for women/men?)

Thank you for this input; we included all the vignettes in the Appendix.

(6) A matter of style: the authors use "we"/"our" too much in the article. When "we"/"our" refers to the research team (e.g. L 41-43) it is acceptable. However when "we"/"our" is used to include also the reader (as in L 34-40 or L 100 "our body") it makes the assumption that there is a consensus regarding the presented issue. I recommend throughout the article the use of a more neutral/scientific style: one, they, it etc.

As requested, we used a more neutral/scientific style throughout the paper.

Reviewer 2 Report

The proposed paper deals with the very important topic of the role of the partner's attractiveness and the partner's comments on body satisfaction and romantic relationship. The use of an experimental approach is extremely commendable. The design included vignettes, but the authors responsibly point out the most important limitations of their usage. Although the paper is well designed and based on a good research design, there are certain doubts and shortcomings that I will mention below. I suggest that the authors include the answers to some of my questions in the text itself (depending on the answer, it may not be necessary to include the answers to some of the questions in the text).

„According to our reasoning, attractive individuals who criticize their mates’ appearance  might produce negative consequences on their partner’s body image and relationship satisfaction.“

I suggest avoiding subjectivity when writing.

„It is our contention that feelings of confidence about either one’s appearance or one’s  dyadic relationship could be decreased in individuals who receive a negative comment from a partner who is seen as highly attractive.“

I suggest “hypothesis” instead of “contention”.

Explain why the opportunistic sampling technique was elected and which adjustments in process of participant recruitment they did during the process.

„Keeping men and women separately, the participants were randomly assigned to one of the four experimental conditions.“

How many participants filled in the questionnaires at the same time?

Comment on the non-uniformity of the sample by BMI - in women's sample the maximum value is at the upper limit of the overweight category, while among men the maximum value is at the lower limit of the extremely obese category. Could this have also resulted in significant effects of BMI covariates for men? I suggest commenting on it in the discussion.

Why was no data collected on dieting as a technique for reducing dissatisfaction with body appearance?

„Moreover, some experimental evidence revealed that even a positive compliment about one’s appearance, which  makes participants feel good in general, might paradoxically lead to increased body shame, as it contributes to focus attention on external [75].“

Is there a possibility that low self-esteem is important to control: people with low self-esteem who receive a compliment that is not in line with their self-image (when they are praised for something they themselves believe they do not have if it is a desirable trait or that they have the opposite an undesirable trait, they will feel worse) will have additional damaged self-esteem?

„Contrary to our prediction, a main effect of partner attractiveness was not found on men’s reported  quality of their relationship.“

How do you explain results that are inconsistent with your hypotheses?

Did you control the participation of both partners in the research? I suggest, if it was not controlled or if both partners were allowed to participate, that the same be commented as a limitation.

Author Response

Reviewer #2:

(1) According to our reasoning, attractive individuals who criticize their mates’ appearance might produce negative consequences on their partner’s body image and relationship satisfaction.” I suggest avoiding subjectivity when writing

As suggested, we avoided subjectivity in our writing; as reported above, we used a more neutral/scientific style throughout the paper.

(2) It is our contention that feelings of confidence about either one’s appearance or one’s dyadic relationship could be decreased in individuals who receive a negative comment from a partner who is seen as highly attractive.”

I suggest “hypothesis” instead of “contention”.

As suggested, we replaced “contention” with “hypothesis”; thank you.

(3) Explain why the opportunistic sampling technique was elected and which adjustments in process of participant recruitment they did during the process.

We explained that “The study participants were recruited using opportunistic sampling techniques, in order to reach the required sample size in a short time and avoid the risk of incurring possible restrictions due to the progress of the COVID-19 pandemic”. We did not make any adjustment in the recruitment process.

(4) Keeping men and women separately, the participants were randomly assigned to one of the four experimental conditions.

How many participants filled in the questionnaires at the same time?

In the Participants and procedure section we specified that “Each participant completed the questionnaire alone and gave it back to the researcher.”

(5) Comment on the non-uniformity of the sample by BMI - in women's sample the maximum value is at the upper limit of the overweight category, while among men the maximum value is at the lower limit of the extremely obese category. Could this have also resulted in significant effects of BMI covariates for men? I suggest commenting on it in the discussion.

Thank you for this comment, which allowed us to improve the Discussion section. As suggested, we reported (p. 9) that “Participants’ BMI was associated with both women’s and men’s body dissatisfaction and body shame, which suggests that thin ideals are internalized by the two genders. Nevertheless, we should notice that among our female participants the maximum BMI value was at the upper limit of the overweight category, while among men the maximum value was at the lower limit of the extremely obese category, which could have affected our results.”

(6) Why was no data collected on dieting as a technique for reducing dissatisfaction with body appearance?

We focused on acceptance of cosmetic surgery, which represented a novel contribution to the literature. Indeed, the association between partner attractiveness and dieting was examined in previous studies (e.g., Reynolds & Meltzer, 2017). Nevertheless, at the end of the Discussion we mentioned that “future studies could examine if partner attractiveness together with appearance-related comments from one’s partner could influence men’s and women’s either positive body image, which is receiving increasing attention in recent years, or some other appearance-enhancing strategies, such as dieting or compulsive exercise”.

(7) Moreover, some experimental evidence revealed that even a positive compliment about one’s appearance, which makes participants feel good in general, might paradoxically lead to increased body shame, as it contributes to focus attention on external [75].“

Is there a possibility that low self-esteem is important to control: people with low self-esteem who receive a compliment that is not in line with their self-image (when they are praised for something they themselves believe they do not have if it is a desirable trait or that they have the opposite an undesirable trait, they will feel worse) will have additional damaged self-esteem?

This was an interesting observation, which we used to propose future research directions; p. 12, “The role of self-esteem could be considered as well, because appearance-related comments could have different effects if they match or mismatch one’s self-view (e.g., positive versus negative comment directed to an individual with high versus low self-esteem).” 

(8) Contrary to our prediction, a main effect of partner attractiveness was not found on men’s reported quality of their relationship.”

How do you explain results that are inconsistent with your hypotheses?

This was a mistake, as we did not hypothesize a main effect of partner attractiveness. We corrected this part deleting the sentence “contrary to our hypothesis”.

(9) Did you control the participation of both partners in the research? I suggest, if it was not controlled or if both partners were allowed to participate, that the same be commented as a limitation.

Thank you for this comment. In the Discussion section we added it as a limitation (“Another limitation is represented by our sampling technique, which was an opportunistic one; moreover, although we did not recruit couples, we did not control if both partners took part in the study; although unlikely, this possibility cannot be ruled out. “
